# Multilocus Sequence Typing and Single Nucleotide Polymorphism Analysis in *Tilletia indica* Isolates Inciting Karnal Bunt of Wheat

**DOI:** 10.3390/jof7020103

**Published:** 2021-02-02

**Authors:** Malkhan Singh Gurjar, Rashmi Aggarwal, Shekhar Jain, Sapna Sharma, Jagmohan Singh, Sangeeta Gupta, Shweta Agarwal, Mahender Singh Saharan

**Affiliations:** 1Fungal Molecular Biology Laboratory, Division of Plant Pathology, ICAR-Indian Agricultural Research Institute, New Delhi 110012, India; head_patho@iari.res.in (R.A.); teenaiari1@gmail.com (S.S.); dhillonjagmohansingh@gmail.com (J.S.); sangeeta0807@gmail.com (S.G.); sinni.vits13@gmail.com (S.A.); mssaharan7@yahoo.co.in (M.S.S.); 2Department of Biotechnology, Mandsaur University, Mandsaur, Madhya Pradesh 45800, India; ainshekhar08@gmail.com

**Keywords:** Karnal bunt, wheat, *Tilletia indica*, multilocus sequence typing, single nucleotide polymorphism

## Abstract

Karnal bunt of wheat is an internationally quarantined disease affecting trade, quality, and production of wheat. During 2015–2016, a severe outbreak of Karnal bunt disease occurred in north-western plain zone of India. The present study was undertaken to decipher genetic variations in Indian isolates of *Tilletia indica* collected from different locations. Seven multilocus sequence fragments were selected to differentiate and characterize these *T. indica* isolates. A phylogenetic tree constructed based on pooled sequences of actin-related protein 2 (ARP2), β-tubulin (TUB), eukaryotic translation initiation factor 3 subunit A (EIF3A), glyceraldehyde-3-phosphate dehydrogenase (GAPDH), histone 2B (H2B), phosphoglycerate kinase (PGK), and serine/threonine-protein kinase (STPK) showed that isolate KB-11 (Kaithal, Haryana) was highly conserved as it was located in cluster 1 and has the maximum sequence similarity with the reference strain. Other isolates in cluster 1 included KB-16 and KB-17, both from Uttar Pradesh, and KB-19 from Haryana. Isolates KB-07 (Jind, Haryana) and KB-18 (Mujaffar Nagar, Uttar Pradesh) were the most diverse and grouped in a subgroup of cluster 2. Maximum numbers of single nucleotide polymorphisms (SNPs) (675) were in the PGK gene across the *T. indica* isolates. The minimum numbers of SNPs (67) were in KB-11 (Kaithal, Haryana), while the maximum number of SNPs (165) was identified in KB-18, followed by 164 SNPs in KB-14. KB-18 isolate was found to be the most diverse amongst all *T. indica* isolates. This first study on multilocus sequence typing (MLST) revealed that the population of *T. indica* was highly diverse.

## 1. Introduction

*Tilletia indica* Mitra is a floret infecting fungal pathogen which causes Karnal bunt disease of wheat. It is a re-emerging disease in India. In 2014–2015, in the north-western plain zone of India, it occurred in epidemic form when up to 15% of disease incidence was reported [1]. The disease was first reported in Karnal district, Haryana, India [2]. Until now, several countries have had restrictions on importing wheat from countries where the disease has been reported [3]. Strict quarantine measures are imposed on wheat-importing countries that have zero tolerance limits [4]. The disease affects the quality and quantity of wheat grain. The fungus has a unique pathogenesis mechanism: it infects the host plants after dikaryotization between compatible mating-type secondary sporidia, colonizing only the endosperm [5,6], and infects the ear in a partial systemic manner [7]. Teliospores survive in the soil for many years in a dormant state, which is the main source of primary inoculum for the disease [8]. These teliospores are liberated during harvesting/threshing to the soil surface, and the disease cycle begins again. The secondary sporidia have been shown to be very durable and can remain dormant and then regenerate very rapidly under favorable conditions [9]. Moreover, the erratic occurrence of Karnal bunt disease is climate-dependent and requires certain conditions, further making it a complex phytopathogen. Seed- or soil-borne teliospores and their successive germination seem to play only a starting role in Karnal bunt epidemics [10].

The disease is difficult to manage effectively using cultural and fungicidal approaches because of its unique infection behavior. The best approach to manage the disease is through the use of resistant cultivars. Phenotyping the wheat genotypes using different pathogenic isolates of *T. indica* does not give a uniform disease reaction pattern as the pathogen is heterothallic, creating much variability. Efforts have been made by previous workers to study pathogenic and genetic diversity based on randomly amplified polymorphic DNA (RAPD)/Universal rice primers (URP)/ inter simple sequence repeats (ISSR) markers and ITS sequences [6,11,12], but genetic variability through multilocus sequence typing (MLST) has not been done. The MLST tool using partial sequence analysis of seven to ten housekeeping genes is the most popular typing approach for epidemiological investigations of phytopathogenic fungi and other microorganisms [13,14]. Among very few studies using MLST in fungal genotyping, multilocus molecular identification of *Colletotrichum tamarilloi,* Damm, P.F. Cannon and Crous revealed mutant isolates and a possible detection of a new pathogenic species affecting tamarillo fruits [15]. In this approach, for each locus studied, different genetic sequences present within a species are assigned as distinct alleles. The combination of the identified alleles at each of the loci defines the allelic profile or sequence type for each isolate. The sequence generated can be used to determine whether the fungal populations are clonal or have undergone recombination patterns. This sequence-based approach produces unambiguous, reproducible results and can be used to compare different isolates of a pathogen. Keeping in view the unique infection biology of the pathogen, and its recent severe outbreak, the study aimed to identify intraspecific variations among twenty *T. indica* isolates using multilocus sequence typing and single nucleotide polymorphisms (SNPs) approaches.

## 2. Materials and Methods

### 2.1. Collection, Isolation and Maintenance of T. indica Isolates

*Tilletia indica* samples were collected from different regions of the north-western plain zone of India from farmers’ field during 2014–2015 where recommended varieties, PBW 343, WH1105 and HD2967, are grown by the farmers. Infected wheat grain samples (Figure 1) brought to the Fungal Molecular Biology Laboratory, Division of Plant Pathology, ICAR-IARI, Pusa Campus, New Delhi, India, were surface sterilized using 70% ethanol. Cultures of *T. indica* isolates were raised from teliospores using the earlier standardized technique by [16]. For the separation of teliospores, bunted grains were vortexed in a sterile, capped vial containing 10 mL sterile distilled water. Each teliospore suspension was filtered through a 45-µm mesh sieve to remove grain chaff, and the filtrate was collected in 15-mL sterile, screw-capped centrifuge tube. To pellet down teliospores, tubes were centrifuged at 10,000 rpm for 2 min, and the supernatant was discarded. The teliospore pellet was re-suspended in 7 mL of 4% sodium hypochlorite and incubated for 2 min for sterilization. After incubation, tubes were centrifuged, the supernatant discarded, and the pellet was washed in a similar way twice using sterile distilled water. Finally, the pellet was re-suspended in 7 mL sterile distilled water and kept overnight in the refrigerator. About 0.5 mL of teliospore suspension was poured on water agar plates (1.5%) and incubated at 16 ± 2 °C in an incubator with exposure to alternate light and dark periods of 12 h. Plates were observed microscopically on a regular basis for the germination of teliospores. A small disc of water agar bearing a single germinating teliospore was transferred to potato-dextrose agar (PDA) test tube slants and incubated at 16 ± 2 °C with alternate light and dark periods. After 3–4 days, a creamy white powdery growth of fungus appeared (Figure 1) showering sporidia downward from the disc, which covered the entire slant within a few days. These *T. indica* isolates were further maintained at 16 ± 2 °C in the incubator by regular sub-culturing.

### 2.2. DNA Isolation

Fungal DNA was isolated from 10–15-day-old mycelium of *T. indica* isolates which were grown on a shaker incubator at 16 ± 2 °C in potato dextrose broth (PDB) using the CTAB (cetyl trimethylammonium bromide) method [17]. Fungal mycelium was collected by filtering from the broth through autoclaved Whatman paper in aseptic conditions and used immediately for isolation of DNA. Approximately 5 g of mycelium was ground in liquid nitrogen in a mortar and transferred to a sterile 30-mL polypropylene tube. In total, 10 mL preheated CTAB extraction buffer (100 mM Tris pH 8.0, 5 M NaCl, 0.5 EDTA (Ethylenediamine tetraacetic acid), pH 8.0, 2% CTAB, 0.1% mercaptoethenol) was added to the ground mycelium powder, mixed gently and incubated at 65 °C for 1 h. After incubation, an equal volume of chloroform–isoamyl alcohol solution (24:1) was added, mixed gently and centrifuged at 10,000 rpm for 10 min. The upper aqueous phase was transferred to a new, pre-sterilized 30-mL polypropylene tube. DNA was precipitated with an equal volume of pre-chilled isopropanol for 2 h at −20 °C. After incubation, DNA was pelleted down by centrifuging at 10,000 rpm for 20 min at 4 °C. DNA pellets were washed twice with 75 % ethanol, air-dried and dissolved in TE buffer (10 mM Tris HCl (pH 8.0) and 1 mM EDTA (pH 8.0)). The concentration of DNA was measured on a nanodrop spectrophotometer (Nanodrop 2000, Thermo Scientific). Qualitative assessment of the DNA was performed by running on 0.8% agarose gel stained with ethidium bromide (0.5 µg/L) in TAE (Tris-acetate-EDTA) buffer (40 Mm Tris-acetate, 1 mM EDTA (pH 8.0)) along with the 1 Kb DNA ladder (MBI, Fermentas). Electrophoresis was carried out in 1× TAE buffer at constant voltage (75 V) for 1 h and DNA bands were observed using a gel documentation system (UVITECH, UK, New Fire Reader).

### 2.3. Selection of MLST Loci and Amplification

For the multilocus sequence typing (MLST), the seven genes selected after BLAST analysis with *T. indica* were: actin-related protein 2 (ARP2; AOA177TN81), β-tubulin (TUB; A0A177TPV8), eukaryotic translation initiation factor 3 subunit A (EIF3A; A0A177TP83), glyceraldehyde-3-phosphate dehydrogenase (GAPDH; A0A177TT56), histone 2 B (H2B; A0A177TGR2), phosphoglycerate kinase (PGK; A0A177TT19) and serine/threonine-protein kinase (STPK; A0A177TLFO). These genes are highly conserved in fungi. Housekeeping genes are typically constitutive genes that are required for the maintenance of basal cellular functions that are essential for the existence of a cell, regardless of its specific role in the tissue or organism. The primers (Table 1) were designed using IDT Oligo Analyzer from sequences taken from NCBI GenBank (http://www.ncbi.nlm.nih.gov/). The designed primers were synthesized by GCC Biotech Pvt. India. The PCR amplifications were carried out in 25µL reaction volume, consisting of 200 ng of genomic DNA, 200 µmol/L dNTP mix (dATP, dGTP, dCTP, dTTP), 0.1 µmol/L each primer, 3.5 mmol/L MgCl_2_, 1.5 U *Taq* DNA polymerase, 9.5 µL water, and 1× *Taq* buffer in a thermal cycler (Bio-Rad Laboratories India Pvt Ltd.). The PCR conditions used were: 95 °C for 5 min—initial denaturation; 95 °C for 1 min—denaturation for 35 cycles at 55–59 °C for 30 s—annealing (gradient); 72 °C for 1 min—extension; and 72 °C for 7 min—final extension with three replications.

The PCR products were separated on 1.2% agarose gel stained with ethidium bromide (0.5 µg/L) in TAE buffer (pH 8.0) along with the 1 Kb DNA ladder (MBI, Fermentas). Electrophoresis was carried out in 1× TAE buffer at constant voltage (75 V) for 1 h. Amplified PCR products were visualized using the gel documentation system.

### 2.4. Sequencing and Phylogenetic Analysis

The amplified PCR products of each gene, viz. actin-related protein 2 (ARP2), β-tubulin (TUB), eukaryotic translation initiation factor 3 subunit A (EIF3A), glyceraldehyde-3-phosphate dehydrogenase (GAPDH), histone 2 B (H2B), phosphoglycerate kinase (PGK), and serine/threonine-protein kinase (STPK) were sequenced by Eurofins Pvt. Ltd. Bangaluru, India. Sequences were trimmed for phylogenetic analysis. These sequences were used to determine a final panel of seven gene fragments that gave the highest discrimination for MLST. The evolutionary history was inferred by using the maximum likelihood method based on the Kimura 2-parameter model. The bootstrap consensus tree inferred from 1000 replicates was taken to represent the evolutionary history of the isolates analyzed. Branches corresponding to partitions reproduced in less than 50% of bootstrap replicates are collapsed. The percentage of replicate trees in which the associated isolates clustered together in the bootstrap test (1000 replicates) is shown next to the branches. Initial tree(s) for the heuristic search were obtained automatically by applying Neighbor-Join and BioNJ algorithms to a matrix of pairwise distances, estimated using the maximum composite likelihood (MCL) approach, and then selecting the topology with superior log-likelihood value. For phylogenetic analysis, 20 amplified nucleotide gene sequences, including the reference sequence (pooled selected gene sequences), were taken. Evolutionary analysis was conducted in CLC Genomics Workbench 9.0 (http://www.clcbio.com) and a phylogenetic tree was constructed [18].

### 2.5. Single Nucleotide Polymorphism Analysis

Single nucleotide polymorphism analysis was conducted with the reference genome (LWDF00000000.1) containing seven gene sequences, namely ENA|OAJ04384|OAJ04384.1 *T. indica* actin-related protein 2, ENA|OAJ05357|OAJ05357.1 *T. indica* tubulin, A|OAJ00852|OAJ00852.1 *T. indica* hypothetical protein, ENA|OAJ06275|OAJ06275.1 *T. indica* glyceraldehyde-3-phosphate dehydrogenase, ENA|OAJ03744|OAJ03744.1 *T. indica* hypothetical protein, ENA|OAJ06449|OAJ06449.1 *T. indica* phosphoglycerate kinase and ENA|OAJ01891 |OAJ01891.1 *T. indica* hypothetical protein. Single nucleotide polymorphisms (SNPs) were identified using the Single Nucleotide Variant tool in CLC Genomics Workbench 9.0.

## 3. Results

### 3.1. PCR Amplification and BLAST Search

PCR amplification using template DNA from each isolate showed a sharp clear band of the size with respect to each specific gene, viz. actin-related protein 2, ARP2 (1200 bp); β-tubulin, TUB (1100 bp); eukaryotic translation initiation factor 3 subunit A, EIF3A (1500 bp); glyceraldehyde-3-phosphate dehydrogenase, GAPDH (800 bp); histone 2 B, H2B (300 bp); phosphoglycerate kinase, PGK (1000 bp); and serine/threonine-protein kinase, STPK (1200 bp). Upon homology analysis using the BLAST tool, amplified sequences showed maximum similarity with the respective reference gene in the whole genome sequence of *T. indica* DAOM236416 (NCBI database). After confirmation by BLAST homology, sequences were submitted to GenBank and accession numbers were obtained (Table 2).

### 3.2. Multilocus Phylogenetic Analysis

A phylogenetic tree was constructed based on the pooled amplified gene sequences, namely ARP2, TUB EIF3A, GAPDH, H2B, PGK and STPK, and with the reference gene sequence. The phylogenetic tree grouped the isolates into two major clusters (Figure 2). The reference sequence was positioned in cluster 1. Isolate KB-11 (Kaithal, Haryana) showed the highest similarity to the reference sequence and is therefore located in cluster 1. Among cluster 2, only two isolates, viz. KB-01(Aligarh, Uttar Pradesh) and KB-06 (Sonipat, Haryana), were grouped in sub cluster a. KB-12 (Ambala, Haryana) is an outlier of sub-cluster b, which exhibited genetic diversity from other isolates (KB-08, KB-15, KB-03, KB-05, KB-02, KB-04) present in the same cluster. KB-09 (Bareilly, Uttar Pradesh) was in the outlier of sub-cluster c, indicating that it shares very limited similarity with KB-10, KB-13 and KB-14 isolates. Isolates KB-07 (Jind, Haryana) and KB-18 (Mujaffar Nagar, Uttar Pradesh) were the most diverse and were grouped in sub-cluster d. In further pooled-gene phylogenetic analysis, *T. indica* isolates clustered with *Tilletia walkeri* Cast l. and Carris but not with *Tilletia caries* (DC.) Tul. and C. Tul. and *Ustilago maydis* (DC.) Corda (Figure 3).

### 3.3. Single Nucleotide Polymorphism (SNP) Analysis

A large number of SNPs were identified after comparison of *T. indica* DAOM, LWDF010001 sequences with the MLST locus sequences (Table 3). Phosphoglycerate kinase gene sequences showed the maximum numbers (675) of SNPs. This was followed by 640 SNPs in actin-related protein across *T. indica* isolates. According to SNPs in each gene, the maximum number of SNPs were identified in actin-related protein gene sequences of both KB-13 (Pant Nagar, Uttar Pradesh) and KB-14 (Sultanpur, Uttar Pradesh) (58 SNPs), while histone 2B (H2B) gene sequences were found most conserved among all the genes since either no SNP or very few SNPs (1–2) were observed in different *T. indica* isolates. Pooled SNP data analysis showed that minimum numbers of SNPs were identified in KB-11 isolate (Kaithal, Haryana), which had 67 SNPs, whereas the highest number of SNPs was found in KB-18 (Mujaffar Nagar, Uttar Pradesh), which had 165 SNPs. Isolate KB-18 was found to be the most diverse among all the *T. indica* isolates studied.

### 3.4. Discussion

India is the second largest producer of wheat in the world. Wheat production has reached 103.60 million tons in the country, and India can export surplus wheat to other countries, but Karnal bunt disease is a major limitation in wheat export, causing huge monetary loss in wheat trade. Considering the international quarantine policy of disease [19] and the unique pathogenesis mechanism of *T. indica*, efforts have been made to generate whole genome data of a virulent isolate (RAKB_UP_1) of *T. indica* (NCBI accession no: MBSW00000000) [20] (Gurjar et al. 2019). Sequence typing methods help in understanding the epidemiology, evolution of pathogens and disease outbreaks [21]. The present study supplemented new data on genotypic variation in *T. indica* isolates collected from the north-western plain zone of India with respect to multilocus sequence typing (MLST) and single nucleotide polymorphisms (SNPs).

MLST is a preferred method for the genotyping of strains of a particular species, and it is more reliable and informative. This method identifies major phylogenetic clades and molecular groups in sub-populations of a species. It has been utilized to identify variations among the isolates of pathogenic microbes, especially in evolution, pathogenesis and ecology [22,23]. In the present study, the phylogenetic analysis, based on seven gene sequences, showed that isolates of *T. indica* did not form clusters according to their geographical preferences. Haryana (KB-07) and Uttar Pradesh (KB18) isolates were most diverse amongst all *T. indica* isolates. According to their geographical preferences, there is a high genetic variation among the isolates of *T. indica*. The pathogenic and genetic variability studies in *T. indica* done earlier were based on RAPD and URP markers [6,24,25]. In *T. indica*–wheat interaction, the dikaryotization of the haploid allantoid sporidia takes place before infection on or within the host tissues. A comparison of nucleotide sequences of microbes and pathogens is the most unambiguous way to differentiate strains/isolates [26]. We, therefore, used a seven gene-based MLST scheme to genotype different isolates of *T. indica*. MLST was earlier successfully used to study epidemiology and population structures in many fungal species [27]. These genes taken for the study are all housekeeping genes, still showing high number of polymorphisms in sequences in this species, and were therefore considered useful for designing the MLST scheme.

Compared to the reference sequences, the highest single nucleotide polymorphism was recorded in the ARP gene (6–58 nucleotides/isolate, 58 nucleotides in KB-13 and KB-14) and PKG gene (0–42 nucleotides/isolate, 42 nucleotides in KB-13 and KB-14). The lowest single nucleotide polymorphism was recorded in the H2B gene (0–2 nucleotides/isolate). Pooled SNP data analysis showed that the maximum and minimum polymorphisms occurred in KB-18 and KB-11 isolates (165 and 67 nucleotides, respectively). Earlier studies reported that isolates of *T. indica* undergo sexual reproduction after teliospore germination when compatible mating-type secondary sporidia fuse to form diakaryon and thus increase chances of variation [28]. In earlier studies, the genetic and pathogenic nature of the fungus was found variable due to the recombination of secondary allantoid sporidia. Two types of secondary sporidia are produced like allantoid sporidia and filiform sporidia, of which only the allantoid type is thought to be able to infect and cause the disease. Allantoid secondary sporidia are ballistospores. *T. indica*, being a heterothallic fungus, demands fusion between secondary sporidia of opposite mating types, resulting in high variation [29]. Therefore, the MLST approach was found useful in understanding genetic variation in *T. indica* and, potentially, in epidemiological investigations. It is simpler, faster, and less expensive than whole-genome sequencing of fungal pathogens [30]. By analyzing SNPs, a diagnostic marker for differentiating isolates revealing intraspecific population diversity of *T. indica* can be developed. This is the first study to reveal information on single nucleotide polymorphism in *T. indica* isolates based on MLST loci.

## 4. Conclusions

MLST has great potential as a tool to interpret the population structures of pathogenic fungi which can be utilized in crop improvement programs. The present study revealed that the population of *T. indica* in the north-western plain zone of India was highly diverse. MLST was found to be useful in understanding genetic variations in *T. indica*. Large numbers of putative SNPs were also detected in *T. indica* isolates. This genetic information will be helpful in understanding the epidemiology and in devising management strategies for Karnal bunt disease.

## Figures and Tables

**Figure 1 jof-07-00103-f001:**
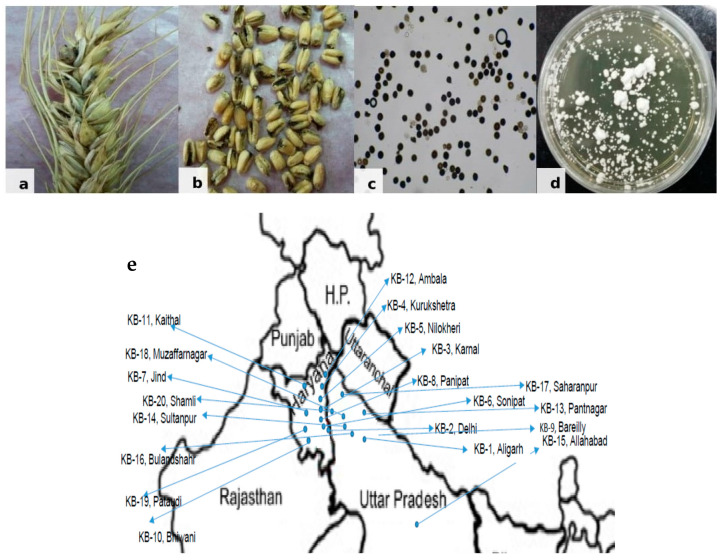
Karnal Bunt disease. (**a**) Infected wheat spike. (**b**) Infected wheat grains. (**c**) Mass of teliospores. (**d**) Mycelial culture of *Tilletia indica.* (**e**) Map showing collected KB samples from north western plain zone of India.

**Figure 2 jof-07-00103-f002:**
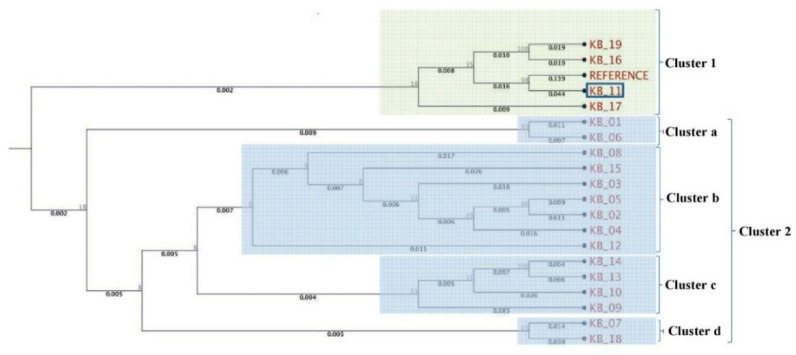
Molecular phylogenetic analysis of *T. indica* isolates using seven gene sequences.

**Figure 3 jof-07-00103-f003:**
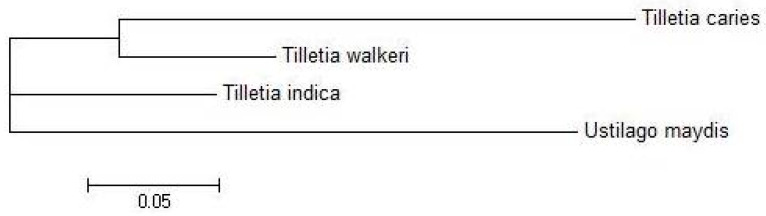
Comparative phylogeny with *Tilletia caries*, *Tilletia walkeri* and *Ustilago maydis*.

**Table 1 jof-07-00103-t001:** List of genes and primers used for multilocus sequence typing in *Tilletia indica*.

S. No.	Genes	Sequences (5′-3′)
1.	actin-related protein 2F actin-related protein 2R	TGCAGCAGCAGGCCAGATCAAG GGAAGGTAGGCCGGGGTACAT
2.	β-tubulin F β-tubulin R	GTCCTTATCGACTTGGAGCCC AGTCCTGGATAGCAGTGCTG
3.	eukaryotic translation initiation factor 3 subunit A F Eukaryotic translation initiation factor 3 subunit A R	TGCACGGTGACCAAGAAACTC CGGAGTTCTTCTTGTCGAGCA
4.	glyceraldeyde-3-phosphate dehydrogenase F glyceraldeyde-3-phosphate dehydrogenase R	TGAAGGGTGGTGCCAAGAAG GAAGACGGAGCTGGCGGAGT
5.	histone 2B F histone 2B R	TCTACAGGCGGCAAGGCTCC GCGGGCAGAGATGGTCGACTT
6.	phosphoglycerate kinase F phosphoglycerate kinase R	GTCCCGATGGTCAGAAGGTCGA AGCATACTCTTCGAGCCCGC
7.	Serine/threonine-protein kinase F Serine/threonine-protein kinase R	CAGATTGTCGCCGCCTGTCAA TTTCTTGCAACAGCCGACGCT

**Table 2 jof-07-00103-t002:** *Tilletia indica* isolates GenBank accession numbers of generated sequences of different genes.

Isolates	Locations	NCBI GeneBank Accession Numbers *						
		ARP2	βTUB	EIF3A	GAPDH	H2B	PGK	STPK
KB-1	Aligarh, Uttar Pradesh	MG655312	MG655332	MG386622	MG386602	MG870326	MG701241	MG655332
KB-2	New Delhi, Delhi	MG655313	MG655333	MG386623	MG386603	MG870327	MG701242	MG655333
KB-3	Karnal, Haryana	MG655314	MG655334	MG386624	MG386604	MG870328	MG701243	MG655334
KB-4	Kurukshetra, Haryana	MG655315	MG655335	MG386625	MG386605	MG870329	MG701244	MG655335
KB-5	Nilokheri, Haryana	MG655316	MG655336	MG386626	MG386606	MG870330	MG701245	MG655336
KB-6	Sonipat, Haryana	MG655317	MG655337	MG386627	MG386607	MG870331	MG701246	MG655337
KB-7	Jind, Haryana	MG655318	MG655338	MG386628	MG386608	MG870332	MG701247	MG655338
KB-8	Panipat, Haryana	MG655319	MG655339	MG386629	MG386609	MG870333	MG701248	MG655339
KB-9	Bareilly, Uttar Pradesh	MG655320	MG655340	MG386630	MG386610	MG870334	MG701249	MG655340
KB-10	Bhiwani, Haryana	MG655321	MG655341	MG386631	MG386611	MG870335	MG701250	MG655341
KB-11	Kaithal, Haryana	MG655322	MG655342	MG386632	MG386612	MG870336	MG701251	MG655342
KB-12	Ambala, Haryana	MG655323	MG655343	MG386633	MG386613	MG870337	MG701252	MG655343
KB-13	Pantnagar, Uttarakhand	MG655324	MG655344	MG386634	MG386614	MG870338	MG701253	MG655344
KB-14	Sultanpur, Uttar Pradesh	MG655325	MG655345	MG386635	MG386615	MG870339	MG701254	MG655345
KB-15	Allahabad, UP	MG655326	MG655346	MG386636	MG386616	MG870340	MG701255	MG655346
KB-16	Bulandshahr, Uttar Pradesh	MG655327	MG655347	MG386637	MG386617	MG870341	MG701256	MG655347
KB-17	Saharanpur, UP	MG655328	MG655348	MG386638	MG386618	MG870342	MG701257	MG655348
KB-18	Mujaffar Nagar, UP	MG655329	MG655349	MG386639	MG386619	MG870343	MG701258	MG655349
KB-19	Pataudi, Haryana	MG655330	MG655350	MG386640	MG386620	MG870344	MG701259	MG655350
KB-20	Shamli, Haryana	MG655331	MG655351	MG386641	MG386621	MG870345	MG701260	MG655351

* PGK = phosphoglycerate kinase, TUB = β-tubulin, ARP2 = actin-related protein 2, GAPDH = glyceraldeyde-3-phosphate dehydrogenase, H2B = histone 2 B, EIF3A = eukaryotic translation initiation factor 3 subunit A and STPK = serine/threonine-protein kinase.

**Table 3 jof-07-00103-t003:** Single nucleotide polymorphism (SNP) with respect to housekeeping genes in different isolates of *Tilletia indica*.

Genes #	ARP2	βTUB	EIF3A	GAPDH	H2B	PGK	STPK	Total Identified SNPs/Isolates
Sequence Length/Reference	1100/1200 bp	960/1350 bp	1400/3300 bp	700/1000 bp	300/430 bp	900/1250 bp	900/1250 bp	
KB-01	42	5	26	8	2	36	9	128
KB-02	28	13	26	9	1	34	7	118
KB-03	7	3	13	8	2	32	3	68
KB-04	22	11	26	9	0	35	10	113
KB-05	29	12	17	10	1	35	14	118
KB-06	40	6	26	10	2	37	14	135
KB-07	41	5	26	22	0	34	6	134
KB-08	23	0	19	5	0	34	16	97
KB-09	37	7	26	14	2	0	23	109
KB-10	6	5	16	6	0	41	17	91
KB-11	7	4	13	10	0	30	3	67
KB-12	38	5	15	10	2	32	20	122
KB-13	58	3	26	10	1	42	13	153
KB-14	58	4	26	8	1	42	25	164
KB-15	7	10	17	22	0	35	14	105
KB-16	41	12	26	18	0	35	17	149
KB-17	40	3	27	9	0	33	8	120
KB-18	38	5	26	43	2	40	11	165
KB-19	40	4	26	17	2	37	17	143
KB-20	38	34	27	7	0	31	8	145
Total	640	151	450	255	18	675	255	

# Where, ARP2 = actin-related protein 2, βTUB = β-tubulin, EIF3A = eukaryotic translation initiation factor 3 subunit A, GAPDH = glyceraldeyde-3-phosphate dehydrogenase, H2B = histone 2 B, PGK = phosphoglycerate kinase and STPK = serine/threonine-protein kinase.

## Data Availability

Data available in a publicly accessible repository. The data presented in this study are openly available in https://www.ncbi.nlm.nih.gov.

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
