# Peer review of "Multilocus Sequence Typing and Single Nucleotide Polymorphism Analysis in Tilletia indica Isolates Inciting Karnal Bunt of Wheat"

_jof, 2021, doi:10.3390/jof7020103_

Round 1
Reviewer 1 Report
This paper describes the isolation and genotyping of the fungi Tilletia indica, a major pathogen of wheat in India, using MLST and SNP identification. The authors isolated DNA, designed primers for 7 "housekeeping genes", conducted PCR, and used phylogenetic tools to find relationships among the fungal isolates.
The methods with conducted using published, sound scientific techniques and the outcome does offer some new insights into the diversity of Tilletia indica in India, but does not have enough novelty to warrant publication. No hypothesis is tested and the authors fall short of discussing the implications of the relationships among the different strains of pathogen isolated from wheat in various regions. There is some mention of this in the discussion (lines 244-245), however, this should be the main focus of the paper. The paper can be redesigned an a few in silico experiments could be done to add to the robustness of the study. Below are some major and minor suggestions for improvement.
Major suggestions:
-The manuscript must have a hypothesis that is being tested. Are you testing whether MLST will work in fungi? Has any other study identified SNPs or MLTS in T. indica? How is your work different from what has been published.
-More in-depth analysis should be done to describe the phylogenetic tree. What does this say about the geographical distribution of the fungi? Are the wheat varieties similar in all locations? Do some T. indica clades/clusters infect specific wheat varieties or are some clades more pathogenic? Does the pathogenicity or level of infection of the wheat (infection rates, endosperm damages, yield losses)?
-Can a PCA or correlation be done between infection rates and the different clades of the phylogenetic trees?
-Explain more how fungal life cycles (dikaryotization, haplotypes, etc) can affect fungal diversity. This is briefly mentioned, but not all readers may be aware of T. indica's alternation of generations.
Minor suggestions:
-Indica in the title should be lower case.
-Section 2.2 on DNA isolation can be shortened significantly. There are numerous papers that describe this method almost verbatim. If there is published study, just cite the source for your DNA isolation and gel electrophoresis protocol and note any medications.
-Lines 137-138. Why were the sequences shortened? Did you eliminate any important nucleotides in the open reading frames or other areas that might have been subject to evolutionary pressure? Can you redo with the entire sequences intact.
Table 1 can be in supplemental information as can most of section 2.5 which could be listed in a tabular format.
Table 3 is very confusing to understand and not all numbers are clearly explained. Can this be listed in a matrix?
Author Response
Reviewer 1
This paper describes the isolation and genotyping of the fungi Tilletia indica, a major pathogen of wheat in India, using MLST and SNP identification. The authors isolated DNA, designed primers for 7 "housekeeping genes", conducted PCR, and used phylogenetic tools to find relationships among the fungal isolates.
The methods with conducted using published, sound scientific techniques and the outcome does offer some new insights into the diversity of Tilletia indica in India, but does not have enough novelty to warrant publication. No hypothesis is tested and the authors fall short of discussing the implications of the relationships among the different strains of pathogen isolated from wheat in various regions. There is some mention of this in the discussion (lines 244-245), however, this should be the main focus of the paper. The paper can be redesigned an a few in silico experiments could be done to add to the robustness of the study. Below are some major and minor suggestions for improvement.
Major suggestions:
-The manuscript must have a hypothesis that is being tested. Are you testing whether MLST will work in fungi? Has any other study identified SNPs or MLST in T. indica? How is your work different from what has been published?
Reply
The hypothesis is genetic recombination or mating behaviour between two compatible monosporidia just before infection. Few reports are studied the MLST in fungi. This is first study using MLST in T. indica. In other studies, SNPs were not identified in T. indica. In our investigation, the present study revealed that the population of T. indica in the North-western plain zone of India. MLST was found to be useful in understanding genetic variation in T. indica. Large numbers of putative SNPs were also detected in T. indica isolates. A valuable option for faster, cheaper and more efficient molecular typing for Karnal Bunt.
-More in-depth analysis should be done to describe the phylogenetic tree. What does this say about the geographical distribution of the fungi? Are the wheat varieties similar in all locations? Do some T. indica clades/clusters infect specific wheat varieties or are some clades more pathogenic? Does the pathogenicity or level of infection of the wheat (infection rates, endosperm damages, yield losses)?
Reply
- indica isolates used in present study belong to two major wheat growing states of India that Haryana and UttarPradesh. Map is showing the collection of diseased samples. In these places, recommended verities, PBW 343, WH1105, and HD2967 are grown by farmers. The pathogenic variability data is not available with us but earlier report revealed that the pathogen is highly variable due to recombination behaviour between compatible sporidia.
-Can a PCA or correlation be done between infection rates and the different clades of the phylogenetic trees?
No. it cannot possible due to lack pathogenic variability data.
-Explain more how fungal life cycles (dikaryotization, haplotypes, etc) can affect fungal diversity. This is briefly mentioned, but not all readers may be aware of T. indica's alternation of generations.
Reply
Two types of secondary sporidia are produced like allantoid sporidia and filiform sporidia of which only the allantoid type is thought to be able to infect and cause the disease. Allantoid secondary sporidia are ballistospores. This has been elaborated in the manuscript.
Reply
Minor suggestions:
-Indica in the title should be lower case.
Reply
Corrected
-Section 2.2 on DNA isolation can be shortened significantly. There are numerous papers that describe this method almost verbatim. If there is published study, just cite the source for your DNA isolation and gel electrophoresis protocol and note any medications.
Reply
Already, DNA isolation protocol is written in brief.
-Lines 137-138. Why were the sequences shortened? Did you eliminate any important nucleotides in the open reading frames or other areas that might have been subject to evolutionary pressure? Can you redo with the entire sequences intact.
Reply
The sequence were trimmed but not from open reading frames. We cannot redo the entire sequences because the sequences were submitted in NCBI database and in public domain.
Table 1 can be in supplemental information as can most of section 2.5 which could be listed in a tabular format.
Reply
Table 1 is an important information related to genes and primers used. Already it placed properly in section 2.3.
Table 3 is very confusing to understand and not all numbers are clearly explained. Can this be listed in a matrix?
Reply
Table 3 is modified properly and adjusted in page. Now all numbers are clearly visible.
Reviewer 2 Report
Dear Authors,
Great job in putting together this manuscript. The methodology you have presented can be a valuable option for faster, cheaper and more efficient molecular typing not only for Karnal Bunt but other pathogens.
Please see below my comments and suggestions.
line 69. Are you able to indicate what wheat varieties were the isolates collected from?
line 90. Can you please add a small map of the North-Western Plains Zone indicating the different regions where the samples were collected. For those not familiar with the regions in India, this will be a good presentation of how close or far the different regions are from each other. This can also further illustrate your results in the phylogenetic analysis which showed that the isolates did not form clusters based on geographical source.
line113. What are the reasons for choosing the seven genes after the BLAST analysis? Please indicate in Section 2.3.
line 192. Add Tilletia indica in the title.
line 206. Please improve entries in Table3. The numbers can be very confusing.
Author Response
Reviewer 2
Great job in putting together this manuscript. The methodology you have presented can be a valuable option for faster, cheaper and more efficient molecular typing not only for Karnal Bunt but other pathogens.
Please see below my comments and suggestions.
line 69. Are you able to indicate what wheat varieties were the isolates collected from?
Reply
KB infected grain samples were collected from farmer’s field of NWPZ of India where recommended verities, PBW 343, WH1105, and HD2967 are grown by farmers.
line 90. Can you please add a small map of the North-Western Plains Zone indicating the different regions where the samples were collected . For those not familiar with the regions in India, this will be a good presentation of how close or far the different regions are from each other. This can also further illustrate your results in the phylogenetic analysis which showed that the isolates did not form clusters based on geographical source.
Reply
North western plain zone of India map added where disease samples were collected for study.
line113. What are the reasons for choosing the seven genes after the BLAST analysis? Please indicate in Section 2.3.
Reply
Selected seven genes are housekeeping genes. These are highly conserved in fungi. Housekeeping genes are typically constitutive genes that are required for the maintenance of basal cellular functions that are essential for the existence of a cell, regardless of its specific role in the tissue or organism. This has inserted in section 2.3.
line 192. Add Tilletia indica in the title.
Reply
It has corrected.
line 206. Please improve entries in Table3. The numbers can be very confusing.
Reply
Table 3 is modified properly and adjusted in page.
Reviewer 3 Report
I have one big issue about the deposit of sequences of this study, the author said he deposits all the sequences in the NCBI gene bank and obtained an accession number for each sequence. I could not find any of these sequences from this study match with the accessions numbers mention in this paper.
the author should approve that all sequences from this study are a deposit in the NCBI gene bank.
Author Response
Reviewer 3
I have one big issue about the deposit of sequences of this study, the author said he deposits all the sequences in the NCBI gene bank and obtained an accession number for each sequence. I could not find any of these sequences from this study match with the accessions numbers mention in this paper.
the author should approve that all sequences from this study are a deposit in the NCBI gene bank.
Reply
These sequences are available in the NCBI database. I have checked in the nucleotide option of NCBI.
Round 2
Reviewer 1 Report
I still think Table 2 can be shortened/modified.
Author Response
There are no comments. Thanks
Reviewer 3 Report
Dear Editor and Auther
I still have my question about the deposit of accession number tabl#2 in the NCBI GenBank?
All the 7 genes that have been used for detection of the fungi and use the sequence after that to be deposit in the gene bank and get accession number
My question is:
Every sequence for each isolate and each dependent gene should have its own accession number, Why all the isolates from no.KB-1 to KB-20 for example ARP2 gene has the same accession number?
Also, i try to find these accession numbers in the NCBI genebank I could not find any of them.

Author Response
There are no comments. Thanks